# Enhanced Efinaconazole Permeation and Activity Against *Trichophyton rubrum* and *Trichophyton mentagrophytes* with a Self-Nanoemulsifying Drug Delivery System

**DOI:** 10.3390/pharmaceutics17091230

**Published:** 2025-09-22

**Authors:** Seo Wan Yun, Jeong Gyun Lee, Chul Ho Kim, Kyeong Soo Kim

**Affiliations:** Department of Pharmaceutical Engineering, Gyeongsang National University, 33 Dongjin-ro, Jinju 52725, Republic of Korea; wyjm2013@naver.com (S.W.Y.); leepipi87@naver.com (J.G.L.)

**Keywords:** efinaconazole, SNEDDS, solubility, permeability, antifungal activity, *Trichophyton rubrum*, *trichophyton mentagrophytes*, SEM, human nails

## Abstract

**Background**: Onychomycosis responds poorly to topical therapy, and efinaconazole (EFN) has low aqueous solubility. **Methods**: This study aimed to develop a 10% *w*/*w* EFN self-nanoemulsifying system (SNEDDS) with improved solubility, permeation, antifungal activity, and stability. Excipients were screened by EFN saturation solubility. An MCT oil/Solutol HS 15/Labrafil M2125 CS SNEDDS (5/75/20, *w*/*w*) was optimized via a pseudo-ternary diagram. Characterization included droplet size, PDI, and zeta potential, morphology, and drug–excipient compatibility. Solubility was measured across pH. Permeation of EFN SNEDDS vs. EFN suspension was tested by Franz diffusion cells. Antifungal activity against *Trichophyton rubrum* and *Trichophyton mentagrophytes* was assessed by paper-disc diffusion, and hyphal damage on human nails was examined by SEM. Stability was studied for six months under room, accelerated, and stress conditions. **Results**: The optimized SNEDDS formed sub-50 nm droplets with low polydispersity and favourable zeta potential. Solubility was maintained across pH, and cumulative permeation increased 13.6-fold versus suspension. Paper-disc assays showed larger inhibition zones at lower EFN doses. SEM on human nails revealed marked hyphal destruction. TEM confirmed spherical nanoemulsion droplets. FT-IR showed no new peaks, supporting compatibility. Particle size, PDI, zeta potential, and drug content remained stable over six months under all storage conditions. **Conclusions**: A 10% *w*/*w* EFN SNEDDS enhanced solubility, transungual permeation, and antifungal efficacy while maintaining robust stability, supporting its potential as an ethanol-free therapy for onychomycosis.

## 1. Introduction

Onychomycosis is a chronic condition commonly referred to as athlete’s foot. It manifests as nail discoloration, thickening and brittleness and affects approximately 19% of the global population [1,2]. It particularly impacts older adults and individuals with underlying conditions such as diabetes or immunosuppression. In older populations, the prevalence of onychomycosis has been reported to exceed 20% in those aged 60 and above and 50% in those aged 70 and above [3,4]. Approximately one-third of diabetic patients are affected by onychomycosis and exhibit increased susceptibility [5,6]. There are also studies reporting that onychomycosis is up to three times more prevalent in immunocompromised individuals and those with HIV/AIDS [7,8].

Among the dermatophytes causing onychomycosis, *Trichophyton rubrum* and *Trichophyton mentagrophytes* account for over 90% [9]. These fungal species invade keratinized nail structures and penetrate deeply into the nail bed and plate [10]. The dense keratinized structure of the nail plate creates a major barrier to antifungal agent penetration and complicates treatment. Furthermore, the presence of fungal biofilms within the nail unit contributes to treatment resistance and recurrence [11,12]. Current treatment options such as systemic oral antifungal therapies are effective but have side effects like hepatotoxicity that limit their use in some patient populations [13]. Topical formulations often show limited efficacy due to inadequate drug penetration and prolonged treatment durations needed for clinical outcomes [14,15]. This ultimately leads to high relapse rates and makes onychomycosis a recurring health challenge [16]. Efinaconazole (EFN) is a broad-spectrum triazole antifungal agent that inhibits lanosterol 14α-demethylase (CYP51) and disrupts ergosterol synthesis in fungal cell membranes [17,18]. It is a crystalline compound (C_18_H_22_F_2_N_4_O; MW 348.39 g/mol) with a melting point of 86–89 °C and is practically insoluble in water. pH of a saturated solution is between 5.5 and 7.5 [19]. Its marketed 10% solution employs an ethanol-based vehicle to improve solubility, yet EFN’s large molecular size hinder permeation through the dense keratinized nail plate, resulting in suboptimal drug retention in deeper nail layers. Moreover, prolonged topical application can provoke local irritation such as dermatitis or erythema due to ethanol’s irritative properties [1,20]. Consequently, an advanced delivery system is required to enhance EFN solubility, nail penetration and overall bioavailability without increasing adverse effects.

Recently, nanoparticle-based drug delivery systems have gained attention as a promising approach to overcome the limitations of conventional onychomycosis treatments [21]. Various nanotechnology-based systems, including microemulsions, nanoemulsions, liposomes, nanocapsules, solid lipid nanoparticles (SLNs), self-microemulsifying drug delivery systems (SMEDDSs), and self-nanoemulsifying drug delivery systems (SNEDDSs), have been explored to improve drug solubility [22,23,24]. In particular, SNEDDSs offer a promising approach to improve drug solubility and permeability [25]. Comprising a uniform and transparent solution of oil, surfactant, and co-surfactant, SNEDDSs can serve as carriers for poorly soluble drugs. In this study, SNEDDS preconcentrate denotes the anhydrous mixture of oil, surfactant, and co-surfactant, whereas SNEDDS nanoemulsion denotes the oil-in-water dispersion formed upon aqueous dilution of the preconcentrate. By encapsulating lipophilic drugs within nanoemulsion droplets, these systems promote effective penetration through keratinized nail layers and enhance therapeutic efficacy. Additionally, SNEDDSs minimize pH-dependent solubility issues during oral administration by reducing the impact of gastrointestinal factors such as digestive enzymes and pH variations. These formulations overcome the limitations of conventional treatments and hold the potential to reduce treatment durations while improving patient adherence [26,27]. Recent transungual nanosystems improved EFN delivery but showed trade-offs: some reported low EFN loading of 5% *w*/*w*, others achieved good stability yet produced mean droplet sizes ≥ 150 nm, and several enhanced permeation using solvent-rich vehicles with high levels of organic solvent [1,28,29]. Against this backdrop, this study aimed to develop a 10% *w*/*w* EFN-loaded SNEDDS that maintains solubilization after aqueous dilution and yields sub-50 nm droplets. The goal was to improve permeation across nail tissue and to enhance antifungal efficacy against *Trichophyton rubrum* and *Trichophyton mentagrophytes* without organic solvents or active enhancements (Figure 1). We characterized droplet size, polydispersity index (PDI) and zeta potential and verified nanoemulsion morphology by transmission electron microscopy (TEM). Drug–excipient compatibility was assessed by Fourier Transform Infrared (FT-IR). Saturated solubility was determined under various pH conditions and membrane permeation was evaluated in Franz diffusion cells. Storage stability was tested under room-temperature, accelerated and stress conditions. Antifungal activity was evaluated by paper disc diffusion and hyphal damage on human nails was examined by scanning electron microscopy (SEM).

## 2. Materials and Methods

### 2.1. Materials

EFN was purchased from Virupaksha Organics Ltd. (Hyderabad, Telangana, India). Cremophor RH 40, Kolliphor P188, Kolliphor P407, Kollisolv P124, PEG 400, and Solutol HS15 were supplied by BASF (Ludwigshafen, Germany). D-α-Tocopherol polyethylene glycol 1000 succinate (TPGS) and Kollisolv MCT 70 oil were supplied by Boryung Pharmaceutical (Seoul, Republic of Korea). Castor oil, Coconut oil, Corn oil, Linseed oil, Oleic acid, Olive oil, Peanut oil, Polysorbate 80, Sesame oil, Span 80, Sunflower oil, potassium hydroxide, potassium dihydrogen phosphate, acetonitrile, and Ethyl alcohol were purchased from Daejung Chemicals (Siheung, Republic of Korea). Capryol 90, Labrafac PG, Labrafil M1944 CS, Labrafil M2125 CS, Lauroglycol 90, Peceol oil, and Transcutol P were obtained from Gattefossé (St. Priest, France). Phosphate-buffered saline (PBS, pH 7.4) was purchased from Welgene (Gyeongsan, Republic of Korea). Sodium carboxymethyl cellulose was purchased from Duksan Pure Chemical (Ansan, Republic of Korea). Potato dextrose broth (PDB) was purchased from Kisanbio (Seoul, Republic of Korea), and Sabouraud’s dextrose agar (SDA) from BD Biosciences (San Jose, CA, USA). Agar was purchased from Samchun Chemicals (Pyeongtaek, Republic of Korea). Glutaraldehyde was purchased from Sigma-Aldrich (St. Louis, MO, USA), and Osmium tetroxide from TCI (Tokyo, Japan). Deionized water was produced using a distillation device in the laboratory, and all other chemicals used were of analytical grade.

### 2.2. HPLC Method for Sample Analysis

EFN in samples was quantified using an HPLC (Agilent 1260 series; Agilent Technologies, Santa Clara, CA, USA) equipped with a UV-Vis detector and a high-pressure gradient pump. The analytical column was a reversed-phase column (ZORBAX SB-C18, 4.6 × 250 mm, 5 μm; Agilent Technologies, Santa Clara, CA, USA). The mobile phase was a mixture of two components: Component A, a mixture of 0.01 M phosphate buffer (pH 5.5) and acetonitrile in a ratio of 75:25 *v*/*v*, and Component B, pure acetonitrile. The optimized ratio of Components A and B was 20:80 *v*/*v*. HPLC analysis was performed with a flow rate of 1 mL/min. The injection volume was 20 μL, and UV detection was monitored at 210 nm, with the column temperature maintained at 30 °C. The phosphate buffer was prepared by dissolving 0.01 M potassium dihydrogen phosphate in water, with the pH adjusted to 5.5 using a 0.05 M potassium hydroxide solution [30]. Data were acquired and processed using OpenLab CDS CS C.01.08 Chemstation software.

### 2.3. Solubility Study

To select suitable components for the development of the SNEDDS formulation, solubility studies were performed with a variety of oils, surfactants, and co-surfactants. Excess amounts of EFN (10 mg for surfactants and both 10 mg and 200 mg for oils) were added to 1 mL of each pure oil or 10% (*w*/*v*) surfactant solution. The mixtures were briefly vortexed and then stored in a shaking water bath at 37 °C and 50 rpm for 5 days. Afterward, each sample was centrifuged at 13,500 rpm for 10 min to remove undissolved EFN. The resulting supernatant was filtered through a 0.45 µm membrane filter, diluted with a diluent (deionized water and acetonitrile, 40:60 *v*/*v*) [30,31,32,33], and then analyzed via HPLC under the conditions described in Section 2.2. Each solubility test was conducted in triplicate to ensure data reliability.

### 2.4. Construction of Pseudo-Ternary Phase Diagram

A pseudo-ternary phase diagram was constructed to identify the regions where self-emulsification occurs upon dilution and gentle agitation. Based on the solubility study results, Kollisolv MCT 70 oil, Solutol HS15, and Labrafil M2125CS were selected as the oil, surfactant, and co-surfactant, respectively. Different ratios of these mixtures (0.2 mL total) were introduced into 300 mL of deionized water in a glass beaker at 37 ± 0.5 °C, and the mixtures were gently agitated using a magnetic stirrer at 300 rpm. The ability of each formulation to emulsify spontaneously was visually observed, and emulsions were characterized as “good” if mixture droplets dispersed easily in water, forming a fine emulsion. If the emulsions showed immediate oil droplet coalescence or creaming, they were categorized as “bad.” The emulsions were further evaluated based on particle size [34,35]. All tests were conducted in triplicate to ensure accuracy.

### 2.5. Preparation of SNEDDS

Based on the solubility study results, Kollisolv MCT 70 oil, Solutol HS15, and Labrafil M2125CS were selected as the oil, surfactant, and co-surfactant, respectively, for the SNEDDS formulation. The components were weighed in a weight ratio of 5:75:20 (oil–surfactant–co-surfactant) and briefly vortexed until a clear mixture was obtained. Next, 100 mg of EFN was weighed and added to the prepared mixture, followed by additional vortexing until the solution became clear. For physicochemical characterization, an aliquot of the SNEDDS preconcentrate was diluted with deionized water to form a nanoemulsion. Droplet size and PDI were measured as described in Section 2.6.

### 2.6. Droplet Size, Polydispersity Index, and Zeta Potential

The droplet size, polydispersity index (PDI), and zeta potential (ZP) of the SNEDDS formulations were measured to assess emulsion characteristics. Each sample (0.2 mL) was added to 300 mL of deionized water, stirred at 37 ± 0.5 °C and 300 rpm for 15 min, and visually observed for emulsification. The emulsions were then diluted tenfold with deionized water before analysis. Measurements were performed using a Zetasizer Nano ZS (Malvern Panalytical, Malvern, UK) equipped with a disposable folded capillary cell (DTS1070). Prepared samples were introduced into the cell and equilibrated at 37 °C for 120 s. The measurements were conducted at a wavelength of 633 nm and a scattering angle of 13° [36,37,38]. All measurements were performed in triplicate.

### 2.7. Fourier Transform Infrared Spectroscopy

Fourier Transform Infrared (FT-IR) spectroscopy was used to assess the compatibility between EFN and the excipients in the SNEDDS formulation. Spectra were recorded with a Spectrum Two^TM^ spectrometer (PerkinElmer, Waltham, MA, USA). Samples were placed at the centre of the sample holder and scanned over a range of 4000 to 500 cm^−1^ at a resolution of 1 cm^−1^ [39].

### 2.8. Transmission Electron Microscopy Analysis

The morphology and particle size of the EFN SNEDDS were analyzed using transmission electron microscopy (TEM, Talos L120C, Thermo Fisher Scientific Inc., Waltham, MA, USA). The SNEDDS was sequentially diluted with deionized water, resulting in a final 40-fold dilution. Approximately 20 µL of the final diluted sample was dropped onto a carbon-coated TEM grid (FCF200-CU-50, Electron Microscopy Sciences, Inc., Hatfield, PA, USA) and dried in a dry oven at 40 °C for 24 h. TEM analysis was performed at an accelerating voltage of 300 kV [40].

### 2.9. Saturated Solubility Study

The saturated solubility of EFN was evaluated in deionized water, with buffer solutions at pH 1.2, pH 4.0, pH 6.8, and pH 7.4. These buffer solutions were prepared using 0.1 M hydrochloric acid and sodium chloride, using 0.05 M sodium acetate buffer, by combining 0.2 M potassium dihydrogen phosphate with 0.2 M sodium hydroxide [41], and using commercially available phosphate-buffered saline, respectively. In total, 10 mg of EFN was added to 1 mL of each solution, and the mixtures were briefly vortexed. The subsequent experimental procedures followed the same methods described in Section 2.3.

### 2.10. In Vitro Artificial Transdermal Membrane Permeability Study

The in vitro transdermal permeability of the EFN SNEDDS was evaluated using a synthetic membrane (Strat-M^®^, EMD Millipore, Temecula, CA, USA) and a Franz diffusion cell system (DHC-6TD, LOGAN Instruments, Somerset, NJ, USA). Each Strat-M^®^ membrane disc was mounted onto the receiver compartment with an effective diffusion area of 1.77 cm^2^. The receiver compartment was filled with 5 mL of phosphate-buffered saline (pH 7.4) containing 1% (*w*/*v*) Tween 80, stirred at 600 rpm, and maintained at 32 ± 0.5 °C. The EFN suspension (in 0.5% (*w*/*v*) CMC Na solution) and EFN SNEDDS, both diluted threefold with deionized water, were loaded into the donor compartment at a volume of 0.35 mL each. After loading, 5 mL samples were collected from the receiver compartment at 0.5, 1, 2, 3, 4, 6, 8, 10, 12, 18, 24, and 48 h. The same volume was immediately replaced with fresh PBS containing 1% Tween 80 to maintain sink conditions. Each collected sample was diluted twofold with acetonitrile, filtered through a 0.45 μm nylon filter and then analyzed for EFN concentration using HPLC [29]. All samples were analyzed in triplicate. The cumulative amount of drug permeated (Q), permeability efficiency at 48 h (Q%_48 h_), steady-state flux (*J*), apparent permeability coefficient (*P_app_*), lag time, and permeability coefficient (K_p_) across the artificial membrane were calculated to evaluate the transdermal permeation of EFN [1,42,43,44]. *Q* (mg/cm^2^) was calculated using the following equation:(1)$$Q=\frac{\sum \left(C_{n}\times V_{n}\right)}{A}$$
where *n* is the sampling time point, *C_n_* is the drug concentration at each time interval, *V_n_* is the volume of the sample at each time interval, and *A* is the effective diffusion area. Q%_48 h_ was calculated as the percentage of actual drug permeation relative to the theoretical maximum permeation at the end of the experiment (48 h). *J* (μg/cm^2^/h) was calculated using the following equation:(2)$$J=\frac{dQ}{A\cdot dt}$$
where *dQ*/*dt* is the rate of drug permeation over time, and *A* is the effective area. *P_app_* (μg/h) was calculated using the following equation:(3)$$P_{app}=\frac{J}{C_{0}}$$
where *C*_0_ is the initial concentration of the drug. The lag time (min) required for the drug to saturate the membrane and reach the receiver phase was calculated from the x-intercept of the linear regression line on the concentration–time plot. K_p_ (cm/h) was calculated by dividing *J* by *C*_0_, and the drug absorption characteristics were assessed using Marzulli’s penetrant rating chart [45,46].

### 2.11. Stability Study

The stability of EFN in the SNEDDS formulation was assessed by storing samples in glass vials under different temperature and humidity conditions: 25 °C/60% RH, 40 °C/75% RH, and 60 °C. Samples were monitored over a 6-month period. Droplet size, polydispersity index (PDI), zeta potential (ZP), and drug content were evaluated to determine stability. Particle characterization was conducted according to the methods described in Section 2.6, while drug content was appropriately diluted with a diluent and analyzed using HPLC [47].

### 2.12. In Vitro Antifungal Activity—Paper Disc Diffusion Test

The in vitro antifungal activity of the EFN SNEDDS was evaluated against *Trichophyton rubrum* KACC 45654 and *Trichophyton mentagrophytes* KACC 45553 using the paper disc diffusion test. Both strains were purchased from the Korean Agricultural Culture Collection (KACC). First, conidia suspensions of each fungal strain were prepared at a concentration of 1 × 10^6^ spores/mL. In total, 100 μL of each suspension was evenly spread on either a PDA or SDA plate (diameter 9 cm) and allowed to dry. PDA was used to culture *T. rubrum*, and SDA was used to culture *T. mentagrophytes*. For sample preparation, the EFN suspension (in 0.5% (*w*/*v*) CMC Na) and EFN SNEDDS were prepared at an initial concentration of 100 mg/mL and then serially diluted with deionized water to obtain various concentrations (10, 1, 0.5, 0.1, 0.05, and 0.01 mg/mL). In total, 50 μL of each diluted solution was loaded onto 8 mm diameter sterile filter paper discs. The prepared discs were placed on the surface of agar plates seeded with fungal spores. As controls, discs containing deionized water, 0.5% (*w*/*v*) CMC Na solution, and SNEDDS vehicle solution were used. Plates were incubated at 28 °C for 5 days, and the antifungal activity was determined by measuring the diameter of the inhibition zones around each disc after the incubation period. All tests were performed in triplicate, and the results were recorded as the average inhibition zone diameters [48,49].

### 2.13. Scanning Electron Microscopy Analysis of Morphological Changes in Fungi After EFN SNEDDS Treatment

#### 2.13.1. Morphological Changes in Fungi on Drug-Containing Media

Morphological changes in fungal hyphae after EFN SNEDDS treatment were observed using field emission scanning electron microscopy (FE-SEM, Tescan-MIRA3; Tescan Korea, Seoul, Republic of Korea). *T. rubrum* and *T. mentagrophytes* conidia suspensions (1 × 10^6^ spores/mL) were evenly spread on PDA and SDA plates, respectively, and incubated at 28 °C for 5 days. After incubation, 5 mm-diameter mycelial plugs were cut from the central area of the plates and transferred onto fresh PDA or SDA plates containing the EFN SNEDDS (1 mg/mL) as the treated group and plain media as the control group. Plates were incubated for an additional 5 days under the same conditions. Following incubation, 5 mm diameter mycelial plugs were collected from the treated zones and processed for SEM analysis [48].

For SEM sample preparation, the collected mycelial plugs were fixed in 2.5% glutaraldehyde for 2 h at 4 °C and washed three times with PBS (pH 7.4). Samples were then post-fixed in 1% osmium tetroxide for 1.5 h at 4 °C, followed by graded ethanol dehydration (30%, 50%, 70%, 80%, 90%, and 100%, with 15 min per step). After drying, the samples were sputter-coated with platinum. Morphological changes were examined under FE-SEM [50,51].

#### 2.13.2. Morphological Changes in Fungi on Human Nails

Human nail samples (approximately 1 mm × 2 mm in size) were sterilized with 70% ethanol and prepared for fungal culture. *T. rubrum* and *T. mentagrophytes* conidia suspensions (1 × 10^6^ spores/mL) were evenly spread onto PDA and SDA plates, respectively. The sterilized nails were placed ventral side down onto the agar plates and incubated at 28 °C for 10 days to allow fungal growth on the nail surface. After incubation, the EFN SNEDDS was applied to the surface of each nail at 5 µL for the 100 mg/mL concentration or 20 µL for the lower concentrations (1, 0.1, 0.05, and 0.01 mg/mL). For controls, 20 µL of 0.5% (*w*/*v*) CMC Na or the SNEDDS vehicle without EFN was applied. The plates were incubated at 28 °C for an additional 24 h. The nail samples were carefully collected using sterilized forceps. Morphological changes in fungal hyphae and spores on the treated nails were analyzed using FE-SEM after processing according to the method described in Section 2.13.1 [52].

### 2.14. Statistical Evaluation

All data were expressed as mean ± standard deviation (SD). Analysis of variance (ANOVA) was conducted using Minitab ver. 19 software (Minitab Inc., State College, PA, USA) to examine differences in the mean and standard deviation between groups. By applying a significance level of *p* < 0.05, the differences between the test groups were confirmed as statistically significant.

## 3. Results

### 3.1. Solubility Study

The selection of appropriate oils, surfactants, and co-surfactants is essential for SNEDDS formulations, as each component plays a critical role in enhancing drug solubility, emulsification efficiency, and stability [36,53]. To develop an EFN SNEDDS, solubility studies were conducted using various oils, surfactants, and co-surfactants, as illustrated in Figure 1. The solubility of EFN was initially evaluated at a concentration of 10 mg/mL in various oils. As shown in Figure 1A, most oils exhibited high solubility. To further assess their solubilizing capacity, the EFN concentration was increased to 200 mg/mL, and the results are depicted in Figure 1B. Among the tested oils, Kollisolv^®^ MCT 70 showed the highest solubility at 144.97 ± 5.49 mg/mL and was selected as the oil phase in the SNEDDS formulation. To enhance emulsion stability, SNEDDS formulations utilize high-HLB surfactants with strong affinity for hydrophilic regions and low-HLB co-surfactants with strong affinity for hydrophobic regions. In this study, surfactants were classified based on their HLB values (Figure 1C) [54]. For surfactants with HLB ≥ 10, Solutol^®^ HS 15 exhibited the highest solubility at 8909.77 ± 1170.93 µg/mL and was selected as the primary surfactant. For co-surfactants with HLB < 10, Labrafil^®^ M2125 CS showed a moderate solubility at 468.14 ± 304.50 µg/mL and was chosen for its ability to stabilize the SNEDDS formulation through its interaction with the surfactant. In summary, Kollisolv^®^ MCT 70, Solutol^®^ HS 15, and Labrafil^®^ M2125 CS were selected as the oil phase, surfactant, and co-surfactant, respectively, based on their superior solubility profiles.

### 3.2. Construction of Pseudo-Ternary Phase Diagrams

Pseudo-ternary phase diagrams were constructed using Kollisolv MCT 70, Solutol HS 15, and Labrafil M2125 CS to optimize the SNEDDS composition. A diagram is shown in Figure 2A, with the blue region representing the nanoemulsion formation area. Formulations with surfactant content lower than the oil exhibited poor emulsification, whereas those with at least 60% (*w*/*w*) surfactant and co-surfactant showed efficient emulsification. The visual comparisons in Figure 2B confirm that the 20:70:10 (oil–surfactant–co-surfactant, *w*/*w*/*w*) ratio produced the clearest nanoemulsion. Further evaluation of the particle size and PDI (Appendix A, Appendix A) revealed that F24 (Kollisolv MCT 70:Solutol HS 15:Labrafil M2125 CS = 5:75:20, *w*/*w*/*w*) exhibited the smallest particle size (31.04 ± 9.56 nm) and the lowest PDI (0.16 ± 0.02). Thus, F24, which indicates superior uniformity, was selected for additional studies with drug-loaded formulations [55]. A small droplet size of approximately 30 nm is particularly advantageous, as smaller particles can enhance cellular interactions, thereby improving the likelihood of penetration [56].

### 3.3. Characterization of EFN SNEDDS

#### 3.3.1. Droplet Size, PDI, and ZP

The droplet size, polydispersity index (PDI), and zeta potential (ZP) of the EFN-loaded and EFN-unloaded SNEDDS were analyzed to evaluate their physical properties. The results are summarized in Table 1. EFN-loaded and unloaded SNEDDS showed similar droplet sizes. Upon loading 100 mg of EFN, the droplet size increased slightly to 31.8 ± 0.8 nm, and the PDI also increased to 0.21 ± 0.01. Additionally, the PDI remained below 0.3, confirming the formulation’s homogeneity and stability [57]. The ZP of the EFN-unloaded SNEDDS was −13.6 ± 0.5 mV, and the EFN-loaded SNEDDS showed a similar value of −12.3 ± 0.9 mV, indicating that the incorporation of EFN did not significantly affect the electrostatic stability of the nanoemulsion. These results indicate that the incorporation of EFN into the SNEDDS formulation caused minimal changes in droplet size, PDI, and ZP, while maintaining its stability and uniformity.

#### 3.3.2. FT-IR Analysis

The compatibility between EFN and the excipients in the SNEDDS formulation was evaluated using FT-IR spectroscopy, as shown in Figure 3. The spectrum of EFN (Figure 3A) displays characteristic peaks at 2985.0 cm^−1^ for aromatic C–H stretching, aliphatic C–H stretching at 2940.7 cm^−1^, cyclic alkene C=C stretching at 1614.5 cm^−1^, benzene ring C=C stretching at 1499.1 cm^−1^, alcohol O-H bending at 1420.1 cm^−1^, C–F stretching at 1384.7 cm^−1^, aromatic amine C-N stretching at 1275.5 cm^−1^, and C–O stretching at 1140.9 cm^−1^. The spectrum of the SNEDDS vehicle (Figure 3B) exhibits characteristic peaks at 2923.0 cm^−1^ and 2855.3 cm^−1^ for C=H band, at 1736.0 cm^−1^ for the carbonyl (C=O) group, at 1458.0 cm^−1^ for CH_2_ stretching, and at 1100.8 cm^−1^ for C–O ether bond. The spectrum of the EFN SNEDDS (Figure 3C) shows the characteristic peaks of the vehicle at 2923.4 cm^−1^, 2855.4 cm^−1^, 1736.2 cm^−1^, and 1100.4 cm^−1^, along with the distinctive peaks of pure EFN at 1614.5 cm^−1^, 1499.1 cm^−1^, 1420.1 cm^−1^, 1384.7 cm^−1^, and 1275.5 cm^−1^. Thus, no new peaks or significant shifts were observed, indicating the absence of chemical interactions between EFN and the excipients [58,59,60].

#### 3.3.3. TEM Analysis

The morphological characteristics of the EFN SNEDDS were evaluated using TEM, as shown in Figure 4. The TEM image revealed spherical nano-sized droplets with a distinct boundary, indicating the successful formation of nanoemulsions. The observed droplets were uniform in shape and size, with diameters of approximately 30 nm, consistent with the results obtained from the droplet size analysis. Thus, the TEM findings corroborate the DLS results and confirm a stable nanoemulsion structure.

### 3.4. Saturated Solubility Study

Drug solubility is a critical parameter influencing absorption and bioavailability [61]. Solubility studies were conducted at pH conditions representing the gastrointestinal tract and skin to evaluate the potential of the EFN SNEDDS for both topical and oral applications (Figure 5). Pure EFN exhibited pH-dependent solubility, with the highest value at pH 1.2 and a significant reduction under neutral and alkaline conditions. In particular, it did not dissolve in D.W. and pH 6.8 and showed a low solubility of 27.1 ± 2.1 µg/mL at pH 7.4. In contrast, the EFN SNEDDS maintained consistently high solubility above 8200 µg/mL across all pH conditions, including pH 7.4, which reflects the natural pH of the skin [62]. This improvement can be attributed to the nanoscale droplet size and the surfactant system in the SNEDDS, which increase the surface area for dissolution and stabilize the drug in a dispersed state, thereby preventing precipitation [63,64]. By addressing the pH-dependent solubility limitations of EFN, this formulation improves absorption and bioavailability, suggesting its potential as an effective drug delivery system [65,66,67].

### 3.5. In Vitro Artificial Transdermal Membrane Permeability Study

An in vitro diffusion study was conducted using a Strat-M^®^ membrane composed of two distinct layers mimicking human epidermis and dermis [68,69]. The cumulative amount of EFN permeated from the EFN suspension and EFN SNEDDS is shown in Figure 6. The cumulative amount of EFN permeated over 48 h was significantly higher for the EFN SNEDDS compared to the EFN suspension (*p* < 0.05). The EFN SNEDDS exhibited slightly higher cumulative drug permeation compared to the suspension during the first 2 h after administration. From 3 h onward, the permeation increased significantly and reached a total of 3.195 ± 0.058 mg/cm^2^ after 48 h. This was approximately 13.6 times higher than the suspension, which achieved only 0.235 ± 0.084 mg/cm^2^. Figure 6B,C illustrate the physical differences between the EFN SNEDDS and the EFN suspension. The SNEDDS formed a clear nanoemulsion, while the suspension showed drug precipitation.

Additionally, differences in various permeation parameters were observed depending on the formulation, and the results are summarized in Table 2. The cumulative permeation of EFN after 48 h (Q%_48 h_) was significantly higher for the SNEDDS (9.69 ± 0.18%) compared to the suspension (0.24 ± 0.09%). This indicates that the SNEDDS achieved higher permeation efficiency than the suspension at the same drug concentration. The steady-state flux (J) of the EFN SNEDDS was calculated as 17.31 ± 0.53 µg/cm^2^/h, which was markedly higher than that of the suspension (0.72 ± 0.30 µg/cm^2^/h). This indicates a higher influx in the SNEDDS compared to the suspension [70,71]. The apparent permeability coefficient (P_app_) for the EFN SNEDDS was determined to be 8.38 × 10^−4^ µg/h, significantly surpassing that of the suspension (3.48 × 10^−5^ µg/h). The enhanced P_app_ reflects the superior ability of the SNEDDS to facilitate drug transport across the membrane [43,70]. The lag time for the EFN SNEDDS (0.021 ± 0.018 min) was shorter than that of the suspension (0.058 ± 0.017 min). This shows that the SNEDDS reduced the time required for EFN to traverse the membrane barrier. The permeability coefficient (K_p_) of the EFN SNEDDS was calculated as 2.47 × 10^−4^ cm/h, significantly higher than that of the suspension (7.34 × 10^−6^ cm/h). Marzulli’s classification showed slow permeability for the suspension and moderate permeability for the SNEDDS (Appendix A) [45,72]. This highlights the superior permeation efficiency of the SNEDDS compared to the suspension. The surfactant and oil in the EFN SNEDDS seen to disrupt the lipid arrangement of the membrane, smoothing the drug’s permeation. Additionally, the nanoscale droplet size enhances permeation by increasing the surface area available for diffusion [42,73].

### 3.6. Stability Study

The stability of the EFN SNEDDS was evaluated under standard (25 °C, 60% RH), accelerated (40 °C, 75% RH), and stress (60 °C) conditions over 6 months to assess changes in droplet size, PDI, ZP, and drug content, as summarized in Table 3. Under standard conditions (25 °C, 60% RH), the droplet size slightly increased from 31.8 ± 0.8 nm to 35.0 ± 0.4 nm over 6 months, while the PDI remained low at 0.22 ± 0.02. The ZP showed a slight shift from −12.3 ± 0.9 mV to −12.6 ± 0.6 mV. The drug content remained nearly unchanged at 100.1 ± 0.1%, confirming good physical and chemical stability under standard conditions. Under accelerated conditions (40 °C, 75% RH), the droplet size increased to 36.4 ± 0.2 nm at 6 months, accompanied by an increase in PDI to 0.25 ± 0.01. However, the nano-sized dispersion remained stable without visible phase separation or drug precipitation. The ZP was measured at −13.5 ± 0.5 mV, with its absolute value showing a slight increase. The drug content was maintained at 99.8 ± 0.2%, indicating stability with minor changes under accelerated conditions. Under stress conditions (60 °C), the droplet size increased to 39.2 ± 0.6 nm, and the PDI rose to 0.26 ± 0.03. The ZP was measured as −16.4 ± 0.3 mV, and the drug content decreased slightly to 99.5 ± 0.1%. Despite these changes, the EFN SNEDDS maintained a nano-droplet size and a PDI below 0.3. The increased absolute ZP value indicates enhanced electrostatic repulsion. Additionally, the drug content remained stable, demonstrating good overall stability under stress conditions. Since the SNEDDS does not contain water in the formulation, it is generally recognized to exhibit higher stability compared to other liquid dosage forms [74,75]. Taken together, these results indicate that the physical properties of the EFN SNEDDS did not change significantly during the 6-month monitoring period, even under stress conditions.

### 3.7. In Vitro Antifungal Activity—Paper Disc Diffusion Test

The antifungal activity of the EFN suspension and EFN SNEDDS was evaluated using the paper disc diffusion test against *T. rubrum* and *T. mentagrophytes*, as summarized in Table 4 and Figure 7 [76]. The control groups treated with D.W. and SNEDDS vehicle showed no effect on hyphae growth. The zone of inhibition for the EFN SNEDDS against *T. rubrum* was significantly larger compared to the EFN suspension at all tested concentrations. At 100 mg/mL, the zone of inhibition for the EFN SNEDDS reached 63.59 ± 0.00 cm^2^, which was more than double that of the suspension (28.97 ± 0.23 cm^2^). Even at lower concentrations of 0.5 mg/mL and 0.1 mg/mL, the EFN SNEDDS exhibited zones of 33.34 ± 0.19 cm^2^ and 12.56 ± 0.27 cm^2^, while the suspension showed only 14.51 ± 0.20 cm^2^ and 0.79 ± 0.33 cm^2^, respectively. No inhibition was detected at 0.05 mg/mL for the EFN suspension, whereas the EFN SNEDDS still demonstrated measurable zones of inhibition (2.83 ± 0.03 cm^2^). Similarly, the EFN SNEDDS exhibited superior antifungal activity against *T. mentagrophytes*. At 100 mg/mL, the zone of inhibition for the EFN SNEDDS was 63.59 ± 0.00 cm^2^, while the suspension achieved 34.45 ± 0.04 cm^2^. The enhanced activity of the EFN SNEDDS was also observed at lower concentrations. At 0.5 mg/mL, the EFN SNEDDS demonstrated a zone of inhibition of 23.75 ± 0.10 cm^2^, while the suspension showed only 5.51 ± 2.00 cm^2^. At 0.1 mg/mL, the EFN SNEDDS demonstrated a zone of 6.30 ± 0.03 cm^2^, which was larger than that observed for the suspension (4.15 ± 0.07 cm^2^). Even at the lowest tested concentration of 0.01 mg/mL, the EFN SNEDDS showed a measurable zone of inhibition with 2.45 ± 0.06 cm^2^. In contrast, the suspension exhibited no inhibition at these concentrations. Thus, the antifungal activity of the EFN SNEDDS was superior against *T. mentagrophytes* compared to *T. rubrum*, demonstrating enhanced efficacy even at lower concentrations.

Figure 7 visually supports these findings, where the EFN SNEDDS formulation formed larger and clearer zones of inhibition compared to the EFN suspension across all tested concentrations. The enhanced antifungal activity of the EFN SNEDDS at low concentrations can be attributed to its nanoscale droplet size, which improves solubility and enables efficient drug delivery into fungal cells. Additionally, the oil and surfactant system in the SNEDDS facilitates drug transfer to fungal cell membranes, exposing the formulation to the ergosterol and resulting in greater growth inhibition compared to the aqueous solution [77,78].

### 3.8. SEM Analysis of Morphological Changes in T. rubrum and T. mentagrophytes After EFN SNEDDS Treatment

#### 3.8.1. Morphological Changes in Fungi on Drug-Containing Media

As shown in Figure 8, SEM was used to analyze the morphological changes in *T. rubrum* and *T. mentagrophytes* following EFN SNEDDS treatment. The control groups of *T. rubrum* (Figure 8A,B) and *T. mentagrophytes* (Figure 8C,D) cultured in drug-free media exhibited intact hyphae with smooth surfaces, uniform thickness, and normal structures. In *T. rubrum*, numerous structurally intact conidia were also observed. In contrast, the treatment groups of *T. rubrum* (Figure 8E,F) and *T. mentagrophytes* (Figure 8G,H) grown on media containing the EFN SNEDDS at 1 mg/mL showed significant structural damage. The hyphae exhibited rough, wrinkled surfaces and irregular thickness, indicating cellular disruption. In *T. rubrum*, holes were also observed in the conidia. Notably, *T. mentagrophytes* exhibited globular swelling, a feature reported to be unique to EFN treatment among various azole antifungal agents [17].

#### 3.8.2. Morphological Changes in Fungi on Human Nails

SEM analysis was performed to examine the morphological changes in *T. rubrum* and *T. mentagrophytes* grown on human nails following treatment with the EFN suspension and EFN SNEDDS. Figure 9 shows the control groups of *T. rubrum* (Figure 9A,B) and *T. mentagrophytes* (Figure 9C,D) cultured in 0.5% CMC Na solution. The hyphae displayed intact structures with smooth surfaces and uniform thickness, indicating normal fungal morphology. Similarly, fungal cultures treated with the SNEDDS vehicle (Figure 9E–H) exhibited no significant morphological changes.

In contrast, morphological damage was observed in the treatment groups after exposure to the EFN suspension and EFN SNEDDS for 24 h (Figure 10). For *T. rubrum*, treatment with the EFN suspension at 0.05 mg/mL (Figure 10A,B) and 1 mg/mL (Figure 10C,D) resulted in slight surface roughness and minimal disruption of hyphal structures. However, treatment with the EFN SNEDDS at the same concentrations (Figure 10E–H) caused more severe damage, including wrinkled surfaces, hyphal shrinkage, and visible fragmentation. Additionally, broken or damaged tips of the conidia were observed. Similarly, *T. mentagrophytes* exposed to EFN suspension at 0.01 mg/mL (Figure 10I,J) showed thin hyphae but remained mostly intact with minimal damage. The suspension-treated group at 1 mg/mL (Figure 10K,L) showed distinct shrinkage and swelling in some hyphae, while most others remained largely intact. In contrast, the EFN SNEDDS-treated groups (Figure 10M–P) demonstrated substantial morphological changes with significant hyphal distortion, severe shrinkage, and fragmentation. Treatment with 1 mg/mL EFN SNEDDS (Figure 10O,P) caused extensive fungal damage, leaving only a small amount of hyphae. The spherical particles observed on the nail surface were identified as microconidia [79,80,81]. Conidia serve as both a means of reproduction and a characteristic response to environmental stress or antifungal treatment [82]. This represents a notable outcome of the morphological alterations induced by EFN SNEDDS treatment. At the same drug concentration, the EFN SNEDDS demonstrated a stronger effect on hyphal destruction in *T. rubrum* and *T. mentagrophytes* compared to the suspension. These results indicate that the EFN SNEDDS exhibits superior antifungal activity due to its enhanced solubility and efficient drug delivery.

Unique morphological changes such as globular swelling observed at low concentrations of EFN SNEDDS treatment warranted further investigation. To better understand its antifungal effects, additional experiments were conducted using higher concentrations of the EFN SNEDDS. Figure 11 presents the morphological changes in *T. rubrum* and *T. mentagrophytes* after treatment with 100 mg/mL of the EFN SNEDDS. The untreated control groups of *T. rubrum* (Figure 11A,B) and *T. mentagrophytes* (Figure 11G,H) grown on nails exhibited well-preserved hyphal structures with uniform thickness and typical fungal morphology. Additionally, numerous normal cylindrical or cigar-shaped macroconidia were observed in *T. rubrum*, indicating active and healthy fungal growth [83].

After 5 h of exposure to the EFN SNEDDS, *T. rubrum* (Figure 11C,D) and *T. mentagrophytes* (Figure 11I,J) exhibited pronounced morphological changes. For *T. rubrum*, most hyphae exhibited severe shrinkage, twisted structures, and irregular thickness. Similarly, *T. mentagrophytes* exhibited visible damage, including contracted hyphae and localized fragmentation. Moreover, hyphal swelling, previously noted with low-concentration EFN SNEDDS treatment, was prominently observed at multiple sites. After 24 h of treatment, the effects on fungal morphology became more pronounced. The hyphae of *T. rubrum* (Figure 11E,F) displayed flattened and crumbled structures with visibly disrupted surfaces. Additionally, prominent bulges and protrusions were also observed among the damaged structures. These hyphal changes were similar to those observed in other antifungal studies reporting inhibited spore germination and severe hyphal damage [84]. In the case of *T. mentagrophytes*, no hyphae could be observed after 24 h of treatment with the EFN SNEDDS (Figure 11K,L). The nail surface was instead covered with spherical particles presumed to be microconidia [85]. This finding demonstrates the potent antifungal effects of the EFN SNEDDS and its ability to induce the complete collapse of fungal hyphae. Overall, these results demonstrate the strong antifungal potential of the EFN SNEDDS by inducing severe structural damage in fungal hyphae and supporting its use as a promising antifungal treatment.

## 4. Conclusions

EFN is a hydrophobic antifungal agent with poor solubility and permeability, which hinders its therapeutic efficacy in onychomycosis treatment. To address these limitations, we developed a 10% *w*/*w* EFN-loaded SNEDDS composed of MCT oil, Solutol HS 15 and Labrafil M2125 CS. The system generated sub-50 nm droplets, as confirmed by DLS and TEM, and FT-IR showed no new peaks, supporting drug–excipient compatibility. Key physicochemical attributes—droplet size, PDI, and zeta potential—remained within narrow ranges. Across performance assays the SNEDDS improved EFN behaviour in vitro. Saturated solubility was maintained at high levels over a range of pH conditions. This maintains EFN in a solubilized state during application and creates a large interfacial area at the hydrated nail surface. The surfactant/co-surfactant system improves wetting of keratin and supports sustained partitioning of EFN into the nail plate. These features are consistent with enhanced permeation in Franz diffusion cells and antifungal effects demonstrated on human nails. Membrane transport through Franz diffusion cells increased 13.6-fold versus suspension. Paper-disc assays demonstrated larger inhibition zones against *T. rubrum* and *T. mentagrophytes* at lower doses, and SEM of human nails revealed pronounced hyphal damage. Stability testing under room temperature, accelerated, and stress conditions preserved droplet characteristics and drug content. Taken together, these findings indicate that an ethanol-free EFN SNEDDS can couple small droplet size with robust stability, improved permeation, and enhanced antifungal activity, supporting its potential as a practical platform for transungual therapy. Liposomes can be destabilized by ethanol and often require high-energy size-reduction steps such as sonication or extrusion [86,87]. SLN systems may undergo polymorphic transitions during storage, leading to drug expulsion [88,89]. In contrast, SNEDDS spontaneously forms small oil-in-water droplets upon aqueous dilution and helps maintain the drug in a solubilized state, which benefits transungual delivery [90]. Future studies could include comparison with the marketed 10% solution and exploration of patient-friendly formats such as solid SNEDDS or nail lacquers.

## Data Availability

The original contributions presented in this study are included in the article/Appendix A. Further inquiries can be directed to the corresponding author(s).

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
