# Peer review of "Enhanced Efinaconazole Permeation and Activity Against Trichophyton rubrum and Trichophyton mentagrophytes with a Self-Nanoemulsifying Drug Delivery System"

_pharmaceutics, 2025, doi:10.3390/pharmaceutics17091230_

Round 1
Reviewer 1 Report
Comments and Suggestions for Authors
The manuscript entitled "Enhanced Efinaconazole Permeation and Activity against Tri-chophyton rubrum and Trichophyton mentagrophytes with a Self-Nanoemulsifying Drug Delivery System" represents a detailed investigation aimed at finding an effective system for enhanced drug delivery of Eficonazole. The authors have described very briefly the initial process of surfactants and cosurfactants scanning and with very details the experimental proofs of the effectiveness of the selected SNEDDS. The results are convincing and the conclusions are supported by the results. I would recommend few improvements and more clarity in some descriptions before the publication:
- Please, define more precisely whether "SNEDDS" is a term referring to the surfactant-oil mixture, or to its emulsion in water. Section "2.5. Preparation of SNEDDS" describes the preparation of surfactant-oil mixture with or without EFN. The last sentence of this section (lines 150-152) states: "The particle size of both EFN-loaded and unloaded SNEDDS formulations was then measured and compared according to the method described in Section 2.6." without a description how these particles are obtained. In fact, the emulsion preparation is described in the following section 2.6, on lines 155 to 158.
- How sensitive is the emulsion preparation to the temperature? 37 ± 5 deg C is a very wide temperature window for the mixing, and I would suppose that the results would be very different when the mixing proceeds at 32 deg C compared to mixing at 42 deg C.
- No need to repeat many times "0.2 μm filtered deionized water", it is enough to define this in the Materials section.
- The TEM image on Fig. 4 is of poor quality and not very convincing, the authors should have better quality images to show.
- Section "3.7. SEM Analysis of Morphological Changes in T. rubrum and T. mentagrophytes after EFN SNEDDS Treatment" should be 3.8
Reviewer 2 Report
Comments and Suggestions for Authors
The current research article focusing on the development of SNEDDS formulations for Enhanced Efinaconazole Permeation and Activity against Trichophyton rubrum and Trichophyton mentagrophytes is interesting and matches the scope of the journal. A few sections of the article needs for be further improved for additional information and better quality. Authors are suggested to address the below comments:
- Please elaborate the abbreviations when used for the first time in the text
- Line 15: Is it “Permeation versus EFN suspension” or “Permeation of EFN SEDDS vs suspension”? Please correct the statement
- Please make a note on the physico-chemical properties of enfinaconazole in the introduction
- Please make a detailed note on the objective of current research
- For drug-excipient compatibility studies at what conditions the samples were stored and for how long?
- How was the ratio of oil / surfactant / co-surfactant established?
- Within saturation solubility study how long the samples vortexed and at what condition?
- Figure 2: The selected ratio of 5:70:20 is not present in figure 2B
Reviewer 3 Report
Comments and Suggestions for Authors This study focused on the clinical pain points in onychomycosis treatment, and the designed 10% w/w EFN-SNEDDS (Efinaconazole-Loaded Self-Nanoemulsifying Drug Delivery System) demonstrated advantages in formulation performance, in vitro activity, and stability. The experimental design was standardized, with complete and mutually supportive data, thereby possessing high academic value and potential application prospects. However, the study has the following limitations:- SNEDDS has been extensively applied in research fields such as transdermal drug delivery. The latest research on SNEDDS for transungual drug delivery could be supplemented in the introduction to highlight the innovativeness of this study.
- The study did not explicitly mention direct comparative data (e.g., transungual efficiency, irritation response) between this SNEDDS and commercially available formulations (such as EFN topical solutions), making it difficult to intuitively demonstrate the magnitude of its advantages over existing clinical medications. It is recommended to supplement comparative experiments with commercially available formulations.
- The discussion on the core mechanism was insufficient. No comparison was made between SNEDDS and other nanocarriers (e.g., liposomes, solid lipid nanoparticles) in terms of EFN transungual efficiency and stability, which fails to highlight the unique advantages of SNEDDS in onychomycosis treatment.
- The TEM images were not clear.
- Some expressions were edundant or imprecise, and there were errors in the formatting of certain tables—for instance, the incorrect notation of "10-6" and "10-5" (should be "10⁻⁶" and "10⁻⁵") in Table 2.
